# A Systematic Review and Meta-Analysis of the Effectiveness of Virtual Reality-Based Rehabilitation Therapy on Reducing the Degree of Pain Experienced by Individuals with Low Back Pain

**DOI:** 10.3390/ijerph20043502

**Published:** 2023-02-16

**Authors:** Taeseok Choi, Seoyoon Heo, Wansuk Choi, Sangbin Lee

**Affiliations:** 1Department of Physical Therapy, Howon University, Gunsan 54058, Republic of Korea; 2Department of Occupational Therapy, Kyungbok University, Namyangju 11138, Republic of Korea; 3Department of Physical Therapy, International University of Korea, Jinju 17731, Republic of Korea; 4Department of Physical Therapy, Namseoul University, Cheonan 31020, Republic of Korea

**Keywords:** virtual reality, low-back pain, head-mounted display, visual analog scale, numerical rating scale

## Abstract

Background: The concept of virtual reality (VR)-based rehabilitation therapy for treating people with low back pain is of growing research interest. However, the effectiveness of such therapy for pain reduction in clinical settings remains controversial. Methods: The present study was conducted according to the reporting guidelines presented in the Preferred Reporting Items for Systematic Reviews and Meta-analyses statement. We searched the PubMed, Embase, CENTRAL, and ProQuest databases for both published and unpublished papers. The Cochrane risk of bias tool (version 2) was used to evaluate the quality of the selected studies. GRADEprofiler software (version 3.6.4) was used to evaluate the level of evidence. We analyzed the included research results using RevMan software (version 5.4.1). Results: We included a total of 11 articles in the systematic review and meta-analysis, with a total of 1761 subjects. Having assessed the quality of these studies, the risk of bias was generally low with high heterogeneity. The results revealed a small to medium effect (standardized mean difference = ±0.37, 95% confidence interval: 0.75 to 0) based on evidence of moderate overall quality. Conclusion: There is evidence that treatment using VR improves patients’ pain. The effect size was small to medium, with the studies presenting evidence of moderate overall quality. VR-based treatment can reduce pain; therefore, it may help in rehabilitation therapy.

## 1. Introduction

The understanding of pain mechanisms has advanced greatly with respect to how a pathological state evolves from the normal state, in which pain helps to protect against injury or is a symptom of illness in tissues, such that the pain becomes a disease itself. However, despite these scientific advances, pain remains very demanding to manage clinically [1]. Opioids are commonly recognized as an effective and essential method for managing pain. Previous studies have revealed the significant drawback of opioid use, which increases the likelihood of developing psychiatric disorders. The long-term use of opioids may lead to a loss of control, opioid dependence, accidental overdose, suicidal attempts, and, ultimately, the failure to alleviate pain [2].

Virtual reality (VR) is a computer-generated environment where three-dimensional orientation and interaction are possible. The environment is generally projected in front of the user’s eyes via an advanced head-mounted display with a wide viewing angle and a motion-tracking system [3].

Distraction is an intervention commonly applied during medical procedures, such as watching movies or listening to music. Distraction techniques withdraw attention from a noxious stimulus by straining a patient’s limited attention [3]. VR systems are a relatively new technology that provide distraction and may be more effective than traditional methods.

There is growing evidence that patient-immersive VR systems could help reduce patient pain during surgery with virtually no side effects; these have emerged as useful tools for a variety of healthcare applications. VR has shown promising results in many clinical trials evaluating its use as a distraction intervention during painful and stressful medical procedures. VR has proven particularly effective in reducing procedural pain and is well-received by patients. However, the existing studies have often been small, and minor, rare side effects have been observed. VR therapy, despite its potential benefits, is not without its limitations. Some common disadvantages of VR therapy include the cost, technical difficulties, motion sickness, limited therapy content, the need for specialized training, and limited research [4].

Non-pharmacological methods are an important option for pain management, especially in minimally invasive procedures. One such method, the distraction method, focuses attention on stimuli other than pain and is used to increase pain tolerance and decrease pain sensitivity. VR equipment has been used recently in the clinical field to distract patients [5].

Conventionally, VR rehabilitation therapy has been extensively utilized to ameliorate cognitive impairments or facilitate physical rehabilitation. Given that the methodology of VR-based rehabilitation training includes reducing pain thresholds through eliciting whole-body movements, this thesis was penned after a thorough examination revealed a substantial body of literature examining its application in patients with back pain.

Researchers have hypothesized that VR creates a non-pharmacological form of pain relief by altering the activity of the complex pain control system, in which nerve fibers traveling to the superior colliculus cause a reaction that induces the eyes and head to turn toward the painful area. The internal pain control system is located in the periaqueductal gray of the brain. However, the effects of VR are still being studied, and its exact mechanism of operation is still controversial [6]. A significant amount of research on new rehabilitation intervention methods using VR has been published in clinical and academic fields, and meta-analyses have been conducted regarding this, but there were no studies specifically focusing on pain relief, which has been a more active area of use recently. As VR-based rehabilitation therapy becomes more common, quantitative verification of its therapeutic effect via the integration of similar studies is required, and meta-analysis is an important and effective method for doing so [7].

The present study assesses the efficacy of VR-based therapy compared with a conventional intervention for pain.

## 2. Materials and Methods

This systematic review and meta-analysis were conducted in accordance with the Preferred Reporting Items for Systematic Reviews and Meta-analyses statement (PRISMA) [8]. This systematic review was registered in PROSPERO with the registration number: CRD42021244051. The present study was carried out by a team of four authors, comprising two doctoral-level physical therapy specialists in their 40s, one doctoral-level rehabilitation science and occupational therapy specialist, and one master’s-level physical therapy specialist in their 40s, from January 2020 to January 2021, through a comprehensive research effort.

### 2.1. Criteria for Considering Studies in This Review

We classified the studies by participant type, including studies on all types of neuralgia and musculoskeletal, visceral, mechanical, and nociceptive pain, including sensory pain dimensions, with patients over 18 years of age. We excluded studies related to psychological problems because many of these have been previously reviewed; the inclusion of psychological problems as a secondary finding in the results of the research was not pursued for the purpose of incorporating it as a collateral outcome.

We included all research that involved any form of immersive or non-immersive VR, including commercial game consoles. The comparison group might have received an alternative intervention or no intervention.

There are several methods for measuring pain; we mainly used the visual analog scale and numerical rating scale (NRS), which are similar evaluation scales, for assessing primary outcome measures. It is difficult to implement an accurate and systematic method for numerically interpreting the pain level experienced by the human body. Under this limitation, these scales were used as subgroup items for multidimensional evaluation [9].

We conducted online searches for studies in previous reviews in December 2020 and updated these in May 2022. We searched the following electronic bibliographic databases: Cochrane Central Register of Controlled Trials (CENTRAL), PubMed, and Embase Ovid.

During this search, we excluded studies that were Ph.D. theses in the fields of physical or occupational therapy. Final decisions on whether to include a study were made via consultation with researchers with more than 10 years of research experience. In the included papers, the underlying definition of VR was the same, although it was sometimes expressed differently (e.g., video games vs. electronic games), and different acronyms were sometimes given for the same evaluation tool (e.g., numeric pain rating scale [NPRS] for the NRS). In cases where an included paper had not been uploaded to the Internet, it was difficult to obtain the original paper; it was then necessary to obtain assistance from the bibliographic information officer of our institution’s library.

We also searched the following electronic gray literature databases: ProQuest (www.proquest.com) and EBSCOhost (www.ebsco.com) (accessed on 1 March 2021).

#### 2.1.1. Study Types

In this study, we included both Randomized Controlled Trials (RCTs) and Quasi-Randomized Controlled Trials (QRCTs) (e.g., assigned based on patient number). Among the 10 studies that were analyzed, 9 were RCTs, and 1 was a QRCT. Additionally, we included a study that compared the efficacy of virtual reality interventions with alternative interventions or no interventions.

#### 2.1.2. Participants

The studies included in our analysis involved all types of musculoskeletal pain, neuralgia, visceral pain, mechanical pain, and nociceptive pain, including sensory pain dimensions in individuals over the age of 18 years. We excluded studies related to psychological problems due to the abundance of previous research in this area.

#### 2.1.3. Interventions

The studies included in this analysis utilized virtual reality interventions, including both immersive and non-immersive virtual reality and game consoles. The comparison group either received alternative interventions or no interventions.

#### 2.1.4. Outcome Measures

There are several methods for measuring pain, but our primary outcome measures were the following: Visual Analog Scale (VAS), Numeric Rating Scale (NRS), and Quality of Recovery-40 (QoR-40).

### 2.2. Search Methods for Study Identification

#### 2.2.1. Electronic Searches

The search for relevant studies was conducted in April 2021 and included searches of the following electronic bibliographic databases: Cochrane Central Register of Controlled Trials (CENTRAL; searched on 1 May 2021), PubMed (searched on 1 May 2021), and Embase Ovid (searched on 1 May 2021). Detailed information on the search strategies were used, including the use of controlled vocabulary thesauruses in the field of life science in the EMBASE-EMTREE database.

#### 2.2.2. Searching Other Resources

In addition to electronic searches, we also searched for relevant grey literature through the ProQuest database (www.proquest.com) (accessed on 1 April 2021).

All other papers excluded from the final meta-analysis and synthesis did not meet the inclusion criteria. Four studies were conducted in the USA [10,11,12,13], with seven from elsewhere: 2 in Korea [14,15], 2 in Australia [16,17], 1 in Turkey [18], 1 in Belgium [19], and 1 in Italy [17]. Most studies were conducted in a primary care setting. Most interventions were delivered in an inpatient setting, although one of the studies delivered the intervention in the participants’ own homes [11]. Most of the trials compared a VR-based intervention with a conventional one. The alternative intervention was often described as therapy using a conventional approach. There was 1 three-armed trial with two comparison interventions [15], 9 two-armed trials [11,12,13,14,16,17], and 1 one-armed trial [20].

### 2.3. Selection of Studies and Data Extraction

In the study selection process, the search was performed by the first author. The second author independently screened the titles and abstracts to assess whether the studies met our predefined inclusion criteria. The authors then considered the full text of potentially relevant articles in 2019, if sufficient information to make a decision was not available, the first author contacted each study’s authors. In the next stage, the first and second authors independently reviewed our correspondence with relevant experts to determine which studies to include. The third author made the final decision on each study. If a study appeared to meet our inclusion criteria but was later excluded, we have stated this, including the reasons for exclusion, in the table characterizing the included studies.

For data extraction and management, the review authors independently extracted data using pre-designed methods for each included study on their setting, inclusion and exclusion criteria, population, participant flow, intervention details, and outcome measures and recorded whether all these data were available. We assessed the methodological quality of each study and resolved any disagreement by discussion or based on the recommendation of the third author. We contacted the study authors via email or social media to request any missing information required for the present review. The extracted outcomes were the visual analog scale and NRS pain scores. All studies reported the primary outcomes as continuous variables. We excluded a total of 29 studies, 16 because it was duplicated, 5 because they did not meet the inclusion criteria, 6 because of their outcomes, and 2 because of their study designs.

The commonly used criteria for evaluating the methodological quality of studies in a meta-analysis include guidelines from the Cochrane Handbook, MOOSE guidelines, and PRISMA guidelines, among others. In this study, the guidelines from the Cochrane Handbook were primarily adopted, although other criteria were also referred to. These guidelines provide guidance on the necessary steps for evaluating the methodological quality of a meta-analysis, including appropriate study selection, data collection, result overview, suitability assessment, and statistical validity testing.

### 2.4. Assessment of Risk of Bias

We independently assessed the risk of bias in the included studies in accordance with the Cochrane Handbook for Systematic Reviews of Interventions [21]. We resolved disagreements via discussion with the third author. We assessed the risk of bias using the The Cochrane Collaboration’s tool for assessing risk of bias in randomized trials (version 2, John Wiley & Sons, Chichester, UK).

We categorized each item as having low, high, or unclear risk and have provided citations from the research report along with our justification for the judgment in the table of risk of bias. We created a table summarizing our results using GRADEpro Guideline Development Tool software (2020). We included in our comparison information on the overall quality of evidence in each study, the magnitude of the intervention effect, and the amount of data available for the outcomes considered.

### 2.5. Measures of Treatment Effect

For the continuous variables, the data were analyzed using the mean and standard deviation, and the mean and 95% confidence interval were calculated for multiple subjects in both the intervention and comparison groups. To synthesize the results across different research designs, we used the standardized mean difference and standard error. We used Comprehensive Meta-Analysis software (2013) for data synthesis. Two review authors classified the outcome measures independently in terms of the areas evaluated. If a study presented more than one outcome measure for the same domain, we included in our analysis the measure most frequently used across all the studies.

For our analysis, we compared the VR-based intervention with a conventional one. Six studies [11,12,13,14,15,16] compared the control groups to VR-based-intervention groups. Alemanno (2019) [17] reported a quasi-experimental design with one arm comparing effects pre- and post-intervention. Garcia and Nambi compared VR-based interventions to control groups who received sham-VR treatments; other research conducted an RCT with three post-intervention arms [18,19].

There is a VR-based yoga program using Wii Fit activities such as deep breathing using PlayStation-based VR [20] that promotes movement and pain distraction; other introduced to be used VR-based rehabilitation systems [21]. The author processing missing data did not receive relevant details from the study authors, and therefore, we did not try to replace any missing values. To assess heterogeneity, we used a random effects model as part of the sensitivity analysis. The heterogeneity index (I^2^) was 85%, indicating that there was high heterogeneity, but its influence on our study’s findings was limited.

The studies’ sections on reporting bias assessment were analyzed thoroughly, as the tendency of inconclusive studies to remain unpublished may affect the results of systematic reviews. We attempted to obtain an outcome protocol to evaluate selective outcome reporting. Another factor that can cause bias is the effect of small studies; we used funnel plots to assess the effectiveness of small studies each time [22], and the authors grouped 10 or more studies for specific, continuous outcomes. We examined the funnel plot for each analysis stage; however, interpreting these was difficult due to the small sample size. Additionally, caution should be exercised in interpreting these results because they do little to indicate publication bias; the number of participants was small, and we did not find many studies that reported negative impacts.

### 2.6. Data Synthesis

For data synthesis, we created a table summarizing the results for each type of relevant primary outcome. We ran a separate meta-analysis for each type of intervention and visualized different comparative studies in forest plots. Meta-analyses were performed using Review Manager software (version 5.4.1; Cochrane Collaboration, 2020, John Wiley & Sons, Chichester, UK). We combined the data in a random-effects meta-analysis because we anticipated that there might be natural heterogeneity among the studies due to differences in their interventions, populations, and implementation strategies. For continuous variables, we used the inverse variance method. For studies involving three or more arms, we only extracted data from the two comparisons most relevant to the main theme and concept. If there was an acceptable level of heterogeneity, we combined the results. As part of the sensitivity analysis, we used both fixed and random effects models. In cases where meta-analysis was not appropriate due to unacceptable heterogeneity, we have presented a descriptive summary of the findings of the study. We used the standardized mean difference to compare measurements produced with different instruments. Since researchers want to compare the effects between two groups of interest in a systematic literature review, RCTs directly comparing two groups are ideal. However, there were many cases where no direct-comparison RCTs existed. It was practically impossible to consider many treatments for the same indication in one clinical trial because the control group of interest tended to differ from country to country. If no direct comparison between the two groups of interest exists, network meta-analysis methods such as indirect comparison and mixed comparison methods can be used. The meta-analysis methods for comparing treatment effects can be divided according to the type and number of comparison groups. If the aim is to compare treatment effects between two groups (the treatment and control groups), and there are sufficient studies performing direct comparisons, the direct comparison method can be considered. Otherwise, the indirect comparison method can be considered. To account for cases where a considered study’s methods partly conformed to an RCT, we designed the present study to include as many of these as possible, thus not limiting the review to only RCTs [23,24,25].

## 3. Results

### 3.1. Search Results

In this study, a comprehensive literature search was conducted to identify relevant studies on the use of Virtual Reality (VR) technology. The databases searched include CENTRAL, PubMed, Embase, and ProQuest. In addition to electronic searches, manual searches were also performed to ensure a comprehensive search process. To refine the search results in PubMed and CENTRAL, the Medical Subject Headings and Text Word indexes were utilized, while the Emtree index was used in the Embase search. The inclusion criteria were applied to the initial pool of studies identified, resulting in the final selection of 11 studies [11,12,13,14,15,16,17,18,19,20,21]. The details of the studies that were excluded from the analysis are documented in Figure 1.

### 3.2. Study Results

An exhaustive search of the available literature revealed 11 studies that fulfilled the predetermined inclusion criteria and involved a cumulative sample size of 1761 participants. In an effort to ensure the relevance and rigor of the data being analyzed, studies pertaining to anxiety or post-traumatic stress disorder were excluded, as well as studies that lacked sufficient data [14,26,27] or were mere reviews [28,29]. The 11 selected studies all reported statistically significant and clinically meaningful results, as depicted in Figure 2.

The aim of the study was to identify relevant studies on the use of virtual reality in the management of pain. To this end, a comprehensive electronic database search was performed in CENTRAL (29 studies), PubMed (69 studies), and Embase (109 studies), yielding a total of 207 studies. The search was performed using MeSH (Medical Subject Headings) and TW (Text word) in PubMed and CENTRAL and emtree in Embase. After removing the articles that did not meet the inclusion criteria, a final total of 10 studies were included in the analysis, with a total of 454 participants.

Most of the included studies were conducted by healthcare professionals, although in some cases, the professional background was not reported. Five of the studies took place in the USA, three in Italy, one in Turkey, one in Australia, and one study did not report the location. The majority of interventions were delivered in an inpatient setting, although one study was delivered in the participant’s own home.

The majority of the trials compared virtual reality intervention with a conventional intervention, and the alternative intervention was often described as therapy using a conventional approach. The extracted outcomes were VAS, NRS, and QoR-40 Pain scores, which were all reported as continuous variables in all 10 studies.

### 3.3. Risk of Bias in the Included Studies

Five studies reported adequate randomization processes [18,19,21], whereas one study’s report of sequence generation raised some concerns [20], and five studies had a high risk of bias due to their lack of random sequence generation [11,12,13,15,17]. We judged the intended interventions to be adequate in six studies [14,16,18,19,20,21], whereas the reporting of intended interventions was unclear in four studies [11,12,13,17], and one study had a high risk of bias due to a lack of adequate intended interventions [16]. From our assessment, there was no indication of incomplete outcome data in any of the studies [11,12,13,14,15,16,17,18,19,20,21]. We judged outcome assessors to have been blinded in all 11 studies [11,12,13,14,15,16,17,18,19,20,21]. We could not decide whether there was a risk of selection bias in the reported result in seven studies [14,15,16,18,19,20,21]. We judged that in the other four studies [11,12,13,17], the selection of the reported results was not clearly described.

### 3.4. Publication Bias

The presence of publication bias was assessed using Funnel Plots and the Trim and Fill method. Funnel Plots were utilized to determine if non-significant or small-scale studies were excluded, with the effect size plotted on the x-axis and the sample size of the study on the y-axis. The Trim and Fill method was employed to estimate the potential number of studies that may have been subject to publication bias; however, this did not have any significant impact on the study outcomes. The analysis of the results was conducted using Comprehensive Meta-Analysis Version 3 software (Figure 3).

## 4. Discussion

### 4.1. Summary of the Main Results

The present review ultimately included a total of 11 studies with a total of 1761 participants; the main results are presented in Figure 4. The studies’ experiments used a variety of commercially available gaming or VR programs, and, except for one study that used a home-based remote rehabilitation approach, all interventions were provided in a hospital or clinic setting. All 11 trials compared a VR-based intervention with conventional therapy. The results of the studies were aggregated, and it was found that virtual reality (VR) interventions may potentially reduce pain compared to conventional interventions. This conclusion is in line with the findings of a prior review. VR is a non-pharmacological complementary strategy with proven beneficial outcomes for treating burn patients, for whom it significantly reduces pain. Additionally, the use of VR is fun and makes the time spent in painful procedures and hospitalization more bearable. The benefits of using VR as non-pharmacological complementary therapy have been demonstrated [30]. The therapeutic benefits of VR therapy for burn patients include, but are not limited to, improved pain management, enhanced physical rehabilitation, reduced anxiety and stress, and increased social engagement and emotional well-being. These advantages can potentially lead to a more positive and efficient healing process for burn patients.

Ding et al. showed that applying VR during the preoperative period reduced postoperative pain more than conventional treatment [31]. Distraction is one of the main mechanisms by which pain reduction is achieved because postoperative pain perception requires attention, and there is a limited amount of working memory. Therefore, distracting one’s attention can reduce the resources available to process post-surgical pain. Additionally, emotional regulation is an important mechanism. Common negative emotions, such as fear and anxiety, in the patient after surgery can make them more sensitive to pain perception and discomfort because such emotions trigger more brain activity in the medial pain system, which is influenced by emotions. VR generally alleviates these negative emotions, reducing the patient’s pain intensity and discomfort after surgery [32]. Moreover, some researchers state that VR is more effective than existing distraction methods because of its immersive nature, given that the patient can actively interact with a vivid virtual environment that theoretically requires more attention [33].

Some studies have found that a VR-based intervention is not more effective than a conventional intervention; this contrasts with the findings from the studies reviewed here. Our meta-analysis revealed important benefits associated with VR-based intervention compared to traditional therapeutic approaches. Previous studies have failed to provide conclusive evidence that immersion-style VR, which has been used as an alternative treatment when chemical therapies and other treatments are no longer effective, significantly reduces pain [34]. Although a general trend toward improved pain scores in the VR-based-treatment group was reported in all studies, some methodological issues and small sample sizes limit the strength of these conclusions. However, previous research has primarily been on proofs of concept, and advances in VR technology may have since improved the feeling of immersion and reduced the risk of cybersickness. Thus, VR technology can be integrated into a holistic and multidimensional treatment modality for patients suffering from pain during medical procedures.

### 4.2. Overall Completeness and Applicability of Evidence

The current study was a comprehensive examination of 11 studies investigating the use of virtual reality (VR) in therapy. The sample sizes of the studies were generally small, which limits the interpretability of the results. Future research should consider reducing the range of participants and providing more detailed information on the interventions used to improve the specificity and clarity of the results. The present study analyzed data on adult participants over the age of 18, but it is important to note that VR systems are becoming increasingly familiar to younger populations who may be more adept at adapting to its use. Further research on the use of VR in therapy for adolescents is necessary. Patient feedback has shown a preference for VR-based training over conventional therapeutic exercises due to its less formal and repetitive nature. VR has been widely used to alleviate social and economic stress and has been deemed more practical compared to traditional approaches. The current study also observed an increasing number of commercial VR programs for rehabilitation purposes, though the effectiveness of game-based interventions is difficult to evaluate due to the diversity of treatment goals and methods. No specific program has been developed for a specific disease, which may hinder the collection of information on specific effects. Additionally, there have been few reported cases of side effects.

The current study utilized both direct and indirect comparisons through mixed treatment comparisons and network meta-analysis to compare multiple treatment groups. Both traditional and Bayesian meta-analyses were utilized to obtain integrated estimates. The study also included studies that did not specifically label themselves as randomized controlled trials [35]. When we considered VR programs designed for rehabilitation purposes, we noticed that the number of commercial game programs for the general public is gradually increasing. However, it is difficult to evaluate the effectiveness of game-based interventions because the treatment goals and methods are diverse. Additionally, because no particular program is used for a specific disease, it may be difficult to obtain information on specific effects. There have been few reported incidents of side effects.

To compare several treatment groups simultaneously, mixed treatment comparisons can be performed, using results from both direct and indirect comparisons. Indirect and mixed comparisons are used in network meta-analysis. Direct and indirect comparisons and network meta-analysis can be divided into traditional and Bayesian meta-analyses according to the approach used to obtain integrated estimates. We even included studies that were not specifically labeled as RCTs if they met our research methodological criteria.

To identify possible biases in the review process, our search process shortlisted many primary studies, from which the final 11 studies were selected. Furthermore, we conducted a literature search covering as many potentially relevant related terms as possible to increase the sensitivity of the various terms used. Despite our comprehensive search strategy, if the abstracts were not published in English, we would not have identified some relevant articles during the search process. Two reviewers independently reviewed the abstracts and extracted data, and a third reviewer resolved disagreements to minimize bias.

### 4.3. Agreement with Other Reviews

The effect size for the primary outcome was small to medium, consistent with previous systematic reviews that assessed the effectiveness of various interventions to improve professional practice and found that VR had an effect compared to conventional interventions [30,31,32,36]. The findings of the present review generally match those of the former reviews. However, we conclude that too few studies have been published so far and that there are different opinions on the effectiveness of VR-based therapy. Most of these differences are due to the exceedingly different inclusion and exclusion criteria applied. In the present review, we excluded studies that considered phantom limb pain, anxiety, and pain dimensions, among others. We think that this diversity of study participants greatly influences heterogeneity. The present review included 11 studies and 1761 participants and found that VR-based interventions were more effective than conventional treatments. However, we think a more careful approach is needed to interpret heterogeneity from a clinical point of view. Many studies on VR have been published in recent years, but diversity is still lacking. As such, it is important to improve the quality of research with continuous updates.

Virtual reality (VR)-based therapy, similar to any other medical intervention, may have certain adverse effects or limitations. Some of the key challenges associated with VR-based therapy include the following:

Cybersickness: This is a form of motion sickness that occurs due to the disconnect between the visual and vestibular systems. Some individuals may experience dizziness, nausea, or headaches during or after a VR session.

Technical difficulties: VR systems can be complex and may require specialized hardware, software, and expertise to use. Technical difficulties such as hardware malfunctions or software crashes can disrupt the therapy session.

Interference with other treatments: VR-based therapy may interfere with other forms of treatment, such as physical therapy, medications, or surgery.

Cost: VR systems and associated hardware can be expensive and may not be accessible to all individuals, especially those in developing countries.

Limited evidence: Although there has been some research into the use of VR for rehabilitation, there is still a limited amount of high-quality evidence available on the long-term efficacy and safety of VR-based therapy.

It is important to note that these limitations and challenges may vary based on the specific VR system and therapy being used. It is also important to consult with a healthcare professional to determine whether VR-based therapy is appropriate for a particular individual [37,38].

VR (Virtual Reality) therapy for pain management is believed to work through a mechanism called “distraction”. When a person is immersed in a VR environment, their attention is redirected from their pain to the virtual environment. This distraction can help to decrease the perception of pain. Additionally, VR therapy has been shown to activate certain regions of the brain, including the periaqueductal gray (PAG), which is involved in the regulation of pain. By stimulating the PAG, VR therapy may help to reduce pain levels by altering pain signals before they reach the conscious mind [39]. It is important to note that the exact mechanisms by which VR therapy reduces pain are not fully understood, and more research is needed to determine the precise biological and neurological processes involved.

## 5. Conclusions 

The results of this systematic review demonstrate that VR-based interventions exhibit moderate-to-strong evidence for reducing pain compared to conventional interventions. The standardized mean difference for the primary outcome measure (pain level) was found to be small to medium in magnitude. These findings are consistent with previous systematic reviews of pain management interventions.

However, it is important to note that the heterogeneity of VR devices and applications used in the studies included in this review may impact the generalizability of the results. The type of head-mounted display, level of immersivity, performance specifications (such as resolution and response speed), and VR application program used varied greatly between studies, making it difficult to determine which specific factors may have contributed to the observed effect. Additionally, the discomfort associated with head-mounted displays, particularly for those with neck issues, and the limited reporting on potential side effects (such as motion sickness, nausea, and headache) warrant further investigation. VR devices utilized for pain management in therapy include virtual reality headsets, handheld controllers, and body tracking sensors. VR systems for pain management and therapy include Oculus Rift, HTC Vive, PlayStation VR, Samsung Gear VR, Google Daydream, Windows Mixed Reality Headsets, open-source VR headset platforms, such as DIYRift and Pygmalion, and other custom-built or specialized VR systems.

In conclusion, while the findings of this review suggest that VR-based interventions may be a promising alternative to conventional pain management techniques, further research is necessary to better understand the various factors contributing to the observed effect and to establish the most effective VR-based treatments for low back pain reduction.

## Figures and Tables

**Figure 1 ijerph-20-03502-f001:**
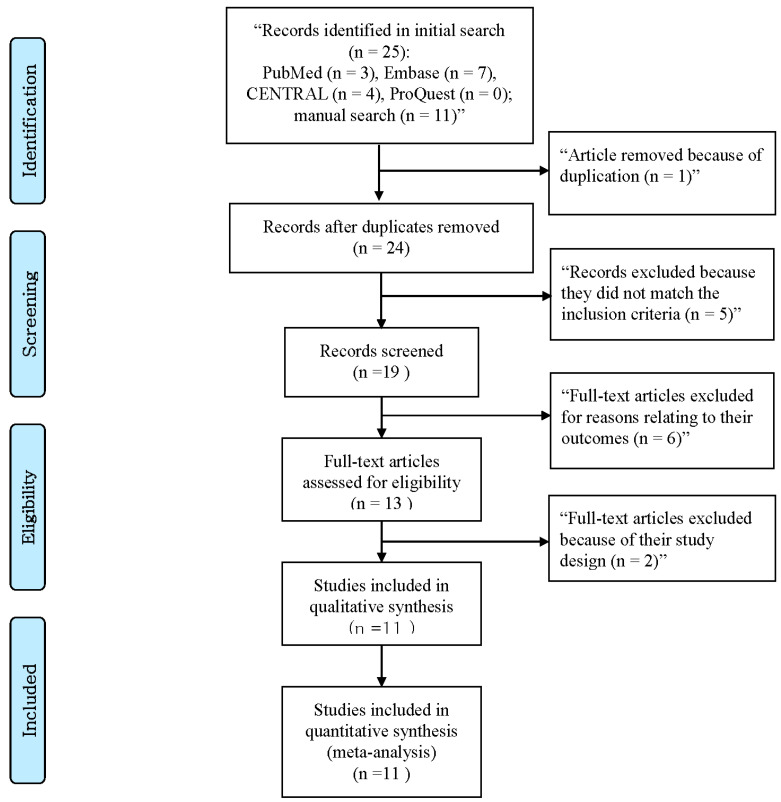
Study flow diagram.

**Figure 2 ijerph-20-03502-f002:**
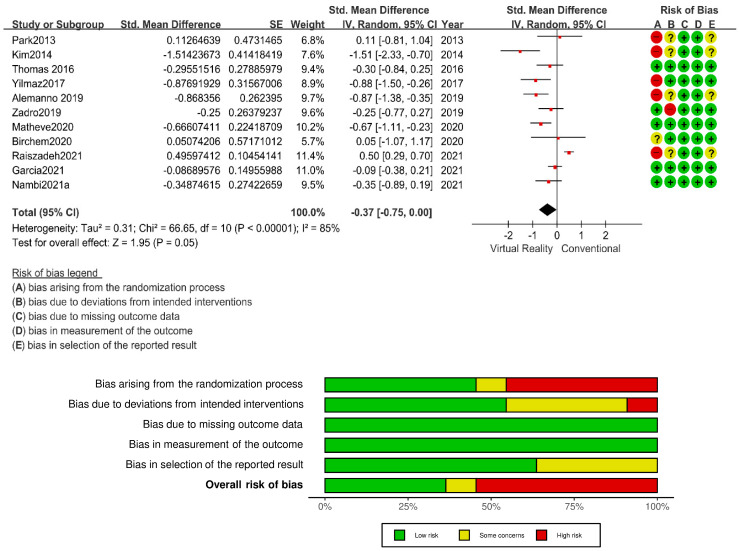
Forest Plot demonstration and Risk of Bias. CI: confidence interval; IV: inverse variance; SE: standard error; Std: standardized [10,11,12,13,14,15,16,17,18,19,20].

**Figure 3 ijerph-20-03502-f003:**
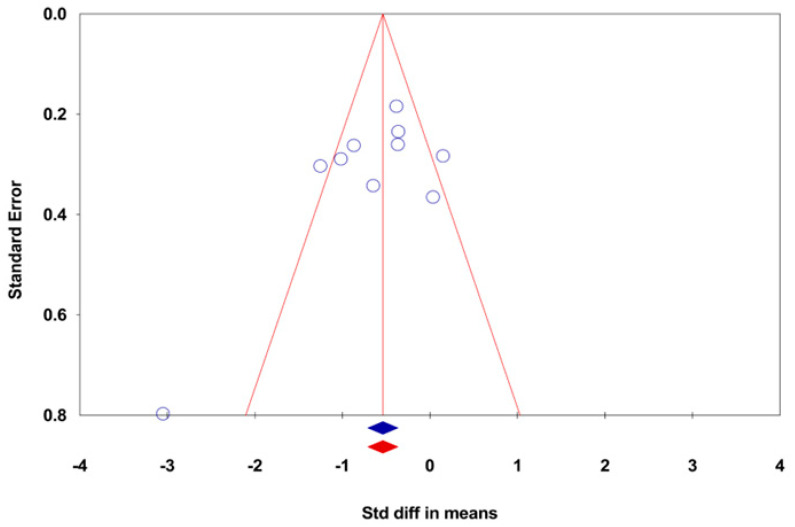
Funnel Plot Representing the Standard Error against the Standardized Difference in Means.

**Figure 4 ijerph-20-03502-f004:**
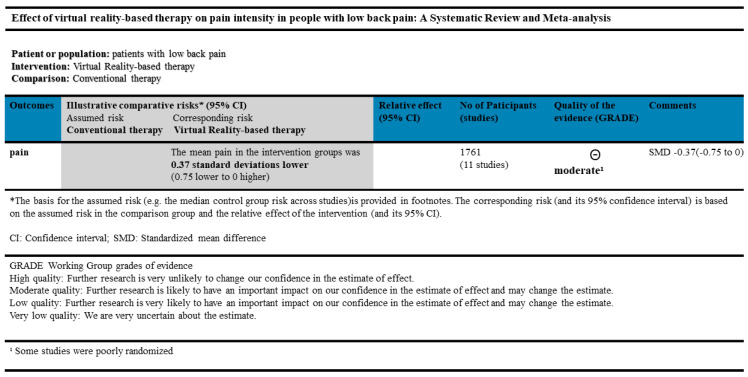
Summary of findings.

## Data Availability

Data available from the corresponding author T.C. on request.

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
