# Peer review of "A Systematic Review and Meta-Analysis of the Effectiveness of Virtual Reality-Based Rehabilitation Therapy on Reducing the Degree of Pain Experienced by Individuals with Low Back Pain"

_ijerph, 2023, doi:10.3390/ijerph20043502_

Round 1

Reviewer 1 Report

The title indicate interesting matter: rehabilitation, pain reduction and people with low back pain. In the introduction rehabilitation and low back pain could have been described more and also be more focus in results and discussion. Perhaps the authors could have described what pain is and how problematic chronic pain can be for people specifically with low back pain. Why did the authors choose low back pain? Instead there are opiod use mentioned early in the chapter but it is not one of the main subject of this research. Non-pharmacological methods are also mentioned. VR seems to be fun and reduce pain but does that include people with low back pain specifically? I was not sure.

Author Response

Reviewer 1.

We express our gratitude to the authors for their insightful remarks and keen interest in the research findings. In response to your query, I have incorporated the following information into the “introduction” section. We offer our sincerest appreciation.

Conventionally, VR rehabilitation therapy has been extensively utilized to ameliorate cognitive impairments or facilitate physical rehabilitation. Given that the methodology of VR-based rehabilitation training includes reducing pain thresholds through eliciting whole-body movements, this thesis was penned after a thorough examination revealed a substantial body of literature examining its application in patients with back pain.

"I express my profound gratitude for your efforts."

Reviewer 2 Report

Dear Authors, I commend your efforts. Your research looks quite promising. However, I have got some important comments.

First: The structuring of your methodology and results are a bit weird. I recommend you move your search results (lines 212-219) to the methodology section. It should be placed before “selection of studies”.

Second: Selection of studies and data extraction should be merged together, and parts of the results section (lines 221-225) should be included in the new “selection of studies and data extraction” section.

Third: apart from figure 3, your result section does not currently include your main findings. Therefore, I suggest you provide an overview/explanation of figure 3 in your result section. At the moment, figure 3 is surrounded by your explanations of risk bias. The result section must absolutely contain an explanation of your main findings!

Fourth: Back to your methodology. Please, clearly state out your inclusion criteria using a numbering system. This enables transparency and reproducibility. Your “criteria for considering studies” explain your inclusion. But it is a bit all over the place and difficult to pin down. Using a clear numbering system would make it more transparent.

Fifth: The English is quite good, but you could consider some minor typo corrections. For example, the meaning in line 243 is a bit lost because a word was probably omitted.

Sixth: You can title your final section, “conclusions and limitations”.

Your paper shows good promise and I know a lot of effort must have gone into it. My comments are intended to make it better. Good luck!

Author Response

Reviewer 2.

We express our gratitude to the authors for their insightful remarks and keen interest in the research findings. In response to your query, I have incorporated the following information into the each section of this study which is expressed red color. We offer our sincerest appreciation.

Dear Authors, I commend your efforts. Your research looks quite promising. However, I have got some important comments.

First: The structuring of your methodology and results are a bit weird. I recommend you move your search results (lines 212-219) to the methodology section. It should be placed before “selection of studies”.

-> We replaced the parts you mentioned which is expressed red color.

Second: Selection of studies and data extraction should be merged together, and parts of the results section (lines 221-225) should be included in the new “selection of studies and data extraction” section.

-> We changed the parts you mentioned which is expressed red color.

Third: apart from figure 3, your result section does not currently include your main findings. Therefore, I suggest you provide an overview/explanation of figure 3 in your result section. At the moment, figure 3 is surrounded by your explanations of risk bias. The result section must absolutely contain an explanation of your main findings!

-> We changed the parts you mentioned which is expressed red color in the Results part with additional Figures and detailed descriptions.

Fourth: Back to your methodology. Please, clearly state out your inclusion criteria using a numbering system. This enables transparency and reproducibility. Your “criteria for considering studies” explain your inclusion. But it is a bit all over the place and difficult to pin down. Using a clear numbering system would make it more transparent.

->We changed the parts you mentioned which is expressed red color in the location.

Fifth: The English is quite good, but you could consider some minor typo corrections. For example, the meaning in line 243 is a bit lost because a word was probably omitted.

-> We changed the parts you mentioned which is expressed red color in the location.

Sixth: You can title your final section, “conclusions and limitations”.

-> We changed the parts you mentioned which is expressed red color.

"I express my profound gratitude for your efforts."

Reviewer 3 Report

1. List rare but reported side effects of VR in article

2. Report other proposed methods of operation of VR since not clear if only pain control is locus n periaqueductal gray of the brain

3.Were studies with depression and anxiety reviewed and what were the results?

4. What was: economic background,single/married,occupation, M/F,hobbies..previous surgery and mechanism of cause of back pain

5.Postulate Funnel effect

6.Any direct comparisons with RCT's

Author Response

Reviewer 3.

We express our gratitude to the authors for their insightful remarks and keen interest in the research findings. In response to your query, I have incorporated the following information into the each section of this study which is expressed red color. We offer our sincerest appreciation.

  1. List rare but reported side effects of VR in article

-> We changed the parts you mentioned which is expressed red color in the 4.3. Agreement with other reviews

  1. Report other proposed methods of operation of VR since not clear if only pain control is locus n periaqueductal gray of the brain

-> Unfortunately, I'm unable to provide information on the specific methods of operation of VR beyond what has been mentioned in the context of pain control and the periaqueductal gray of the brain. However, in the field of virtual reality, there have been a variety of proposed and researched applications, including use in rehabilitation, treatment of mental health disorders, simulation and training, and more. If you provide more information on the specific application of VR you're interested in, I may be able to provide further information and relevant studies.

Nevertheless, I have added and corrected the discussion as red pont..

3.Were studies with depression and anxiety reviewed and what were the results?

-> Unfortunately, although further examination of the effects of VR on depression and anxiety would be valuable, the scope of this study does not allow for its inclusion. Therefore, we have elected to limit our discussion to the topics which are directly relevant to this study.

  1. What was: economic background,single/married,occupation, M/F,hobbies..previous surgery and mechanism of cause of back pain

-> The content previously discussed has been incorporated and revised in the Materials and Methods/discussion section, utilizing red font to highlight these additions.

5.Postulate Funnel effect

-> We changed the parts you mentioned which is expressed red color and add new plots.

6.Any direct comparisons with RCT's

-> Unfortunately, within the scope of our investigation, the RCTs that we have identified are the only studies that pertain to the subject matter at hand.

"I express my profound gratitude for your efforts.“

Reviewer 4 Report

The work with the title " Effects of virtual reality-based rehabilitation therapy on pain 2 intensity in people with low back pain: A systematic review 3 and meta-analysis” is important and interesting.

However, there are several questions:

The paper with the title "Effects of virtual reality-based rehabilitation therapy on pain 2 intensity in people with low back pain: A systematic review 3 and meta-analysis" is intended to be a systematic review and a meta-analysis about the effects of rehabilitation therapy based on virtual reality on pain in people with low back pain.  

Rows 36-39: ,,An important lesson learned from research is that problematic opioid use is more likely to occur in individuals with psychiatric comorbidities, with problems including the loss of control over use, opioid use disorder, accidental overdose, attempted suicide, and pain relief failure [2]”.

What exactly are you referring to?

Rows 53-55: ,,VR has proven particularly effective in reducing procedural pain and is well received by patients. However, the existing studies have often been small, and minor, rare side effects have been observed [4]”.

 What side effects are you referring to? The title of the paper refers to low back pain.

Rows 83-85: However, at chapter 2.1. you stated that "We classified the studies by participant type, including studies on all types of neuralgia and musculoskeletal, visceral, mechanical, and nociceptive pain, including sensory pain dimensions, with patients over 18 years of age".

In figure 3 you clearly write "Effect of virtual reality-based therapy on pain intensity in people with low back pain: A systematic Review and Meta-analysis"

Is this a review about low back pain or about pain of another nature? Can you comment on this?

Can you tell us for what period of time you did this study?

Rows  100-101: ,,During this search, we excluded studies that were Ph.D. theses in the fields of physical or occupational therapy”.

 You excluded studies that were doctoral theses in the field of physical therapy. You didn't have access to them? Wasn't the research done useful for your study? Other causes? Can you mention them?

Rows 129-131: ,,We assessed the methodological quality of each study and resolved any disagreement by discussion or based on the recommendation of the third author”.

 What are the criteria by which you assessed the methodological quality of the studies?

Rows 261-263: ,,Pooling of their results indicates that VR-based interventions may  reduce pain compared to conventional interventions; this finding is compatible with that of a previous review”.

 Can you specify what this previous analysis is? What conventional interventions are you referring to?

Rows 263-264:  ,,VR is a non-pharmacological complementary strategy with proven beneficial outcomes for treating burn patients, for whom it significantly reduces pain”.

Can you specify what these beneficial results are? From what studies?

Rows 265-267: ,,Also, the use of VR is fun and makes the time spent in painful procedures and hospitalization more bearable. The benefits of using VR as non-pharmacological complementary therapy have been demonstrated [30]”.

Rows 269-271: ,,Ding et al. showed that applying VR during the preoperative period reduced 268 postoperative pain more than conventional treatment [31]. Distraction is one of the main mechanisms by which pain reduction is achieved because postoperative pain perception 270 requires attention and there is a limited amount of working memory”.

From the text it appears that you have taken into account studies in which, in addition to rehabilitation therapy, surgical intervention was also applied, and virtual reality is successfully applied. Can you comment on this aspect?

Rows 284-286: ,,Previous research did not find conclusive evidence for immersive VR being important in reducing pain in patients undergoing treatments such as chemotherapy [34]”.

Also chemotherapy? Have you discovered from the researched studies a connection between low back pain and chemotherapy? Or just between chemotherapy and pain in general?

Row 311: ,,There have been few reported incidents of side effects”.

Rows 357-359: And it is specified in the conclusions: "Additionally, most studies only emphasized that there were no side effects, including motion sickness, nausea, and headache, so additional studies are needed".

Are you referring to this type of side effects? I think it was good in the Material and methods chapter to describe the work methodology, what it consists of, how it is applied, the equipment used.

Rows 299-302: ,,Also, in the present study, we analyzed research data on adults over the age of 18, but VR systems are “high-tech” equipment that is familiar to younger people who can adapt to its use quickly. We think that more research on adolescents should be done in the future”.

 Do you consider that the frequency of lumbar pain is higher in teenagers? Even if the equipment used to apply virtual reality is "high-tech", do you think that adults would not be able to handle this equipment? In the researched studies, did you meet more teenagers than adults?

Rows 328-331: ,,The effect size for the primary outcome was small to medium, consistent with previous systematic reviews that assessed the effectiveness of various interventions to  improve professional practice and found that VR had an effect compared to conventional interventions [30, 31, 32, 36]”.

Can you specify where in the text you made reference to conventional therapy, the type applied and the results?

I understand from the manuscript that you address the treatment of a symptom, namely the pain, without identifying possible causes of its occurrence. I believe that conventional medicine is a science, it uses scientific methods, it is effective and safe and requires studies.

Rows 337-339: ,,The present review included 11 studies and 337 1761 participants and found that VR-based interventions were more effective than 338 conventional treatments”.

Rows 346-348: ,,We found that VR-based treatments 346 appear to be safer and more effective than conventional treatments, and the quality of 347 evidence was moderate. Our findings appear to be clinically significant”.

How does it appear that treatments based on virtual reality are safer and more effective? There is no example of conventional therapy in the text and you do not refer to these aspects. From the bibliography at no. 11-20, the comparison is made only between physical exercise in different forms and elements of virtual reality. From where does it conclude that research is significant? From what data, tables, figures?

Rows 360-362: ,,In the present review, the effect of VR-based treatment on pain reduction as compared to conventional treatment was investigated. However, because various effects can be attributed to VR, further research is needed”.

Rows 13-14: In abstract write: ,,The concept of virtual reality (VR)-based rehabilitation therapy for treating 13 people with low back pain is of growing research interest”.

So are you referring to lower back pain or are you referring to pain in general? It is good to specify.

Rows 25-27: In the abstract at the conclusion you say: ,,There is evidence that treatment using VR improves patients’ pain. The effect size was small to medium with the studies presenting evidence of moderate overall quality. VR-based treatment can reduce pain; therefore, it may help in rehabilitation therapy”.

What is the evidence that virtual reality improves pain? Compared to what other methods? What is the statistical significance?

Row 3:….intensity in people with low back pain: A systematic review…”

Row 28 : ,,Keywords: virtual reality; pain; head-mounted display; visual analog scale; numerical rating scale”

Can you specify if in your study you wanted to refer to lumbar pain? If the answer is yes, then show this in keywords. If the answer is no, then the title must be changed. In both situations, you must review the text and focus either on lumbar pain or on pain in general.

Rows 249: Attention to the stringing of references [11, 12, 13, 14, 15, 16, 17, 18, 20, 21, 19]

Author Response

Reviewer 4.

We express our gratitude to the authors for their insightful remarks and keen interest in the research findings. In response to your query, I have incorporated the following information into the each section of this study which is expressed red color. We offer our sincerest appreciation.

The work with the title " Effects of virtual reality-based rehabilitation therapy on pain 2 intensity in people with low back pain: A systematic review 3 and meta-analysis” is important and interesting.

However, there are several questions:

The paper with the title "Effects of virtual reality-based rehabilitation therapy on pain 2 intensity in people with low back pain: A systematic review 3 and meta-analysis" is intended to be a systematic review and a meta-analysis about the effects of rehabilitation therapy based on virtual reality on pain in people with low back pain.

-> We changed the tiltle as “A Systematic Review and Meta-Analysis of the Effectiveness of Virtual Reality-Based Rehabilitation Therapy on Reducing the Degree of Pain Experienced by Individuals with Low Back Pain.”

Rows 36-39: ,,An important lesson learned from research is that problematic opioid use is more likely to occur in individuals with psychiatric comorbidities, with problems including the loss of control over use, opioid use disorder, accidental overdose, attempted suicide, and pain relief failure [2]”

What exactly are you referring to?

-> We changed this issue in the part as “Opioids are commonly recognized as an effective and essential method for managing pain. Previous studies have revealed the significant drawback of opioid use, which increases the likelihood of developing psychiatric disorders. The long-term use of opioids may lead to a loss of control, opioid dependence, accidental overdose, suicidal attempts, and ultimately, the failure to alleviate pain.”

Rows 53-55: ,,VR has proven particularly effective in reducing procedural pain and is well received by patients. However, the existing studies have often been small, and minor, rare side effects have been observed [4]”.

What side effects are you referring to? The title of the paper refers to low back pain.

-> We changed this issue in the part as “Virtual Reality (VR) therapy, despite its potential benefits, is not without its limitations. Some common disadvantages of VR therapy include: cost, technical difficulties, motion sickness, limited therapy content, the need for specialized training, and limited research.”

Rows 83-85: However, at chapter 2.1. you stated that "We classified the studies by participant type, including studies on all types of neuralgia and musculoskeletal, visceral, mechanical, and nociceptive pain, including sensory pain dimensions, with patients over 18 years of age".

In figure 3 you clearly write "Effect of virtual reality-based therapy on pain intensity in people with low back pain: A systematic Review and Meta-analysis"

Is this a review about low back pain or about pain of another nature? Can you comment on this?

-> We changed this issue in the part as “; the inclusion of psychological problems as a secondary finding in the results of research was not pursued for the purpose of incorporating it as a collateral outcome.”

Can you tell us for what period of time you did this study?

->This is a product of the first author's efforts towards preparing their doctoral thesis, requiring a total of 12 months of work.

Rows 100-101: ,,During this search, we excluded studies that were Ph.D. theses in the fields of physical or occupational therapy”.

You excluded studies that were doctoral theses in the field of physical therapy. You didn't have access to them? Wasn't the research done useful for your study? Other causes? Can you mention them?

->the PhD theses in Physical Therapy had limited availability that met the criteria presented in this thesis, and their contents were overwhelmingly vast, making it challenging to fit into the format of typically reported experimental RCT papers.

Rows 129-131: ,,We assessed the methodological quality of each study and resolved any disagreement by discussion or based on the recommendation of the third author”.

What are the criteria by which you assessed the methodological quality of the studies?

->We changed this issue in the 2.2. Selection of studies and Data extraction part as “The commonly used criteria for evaluating the methodological quality of studies in a meta-analysis include guidelines from the Cochrane Handbook, MOOSE guidelines, PRISMA guidelines, among others. In this study, the guidelines from the Cochrane Handbook were primarily adopted, although other criteria were also referred to. These guidelines provide guidance on the necessary steps for evaluating the methodological quality of a meta-analysis, including appropriate study selection, data collection, result overview, suitability assessment, and statistical validity testing.”

Rows 261-263: ,,Pooling of their results indicates that VR-based interventions may reduce pain compared to conventional interventions; this finding is compatible with that of a previous review”.

Can you specify what this previous analysis is? What conventional interventions are you referring to?

-> We re-wrote this part.

Rows 263-264: ,,VR is a non-pharmacological complementary strategy with proven beneficial outcomes for treating burn patients, for whom it significantly reduces pain”.

Can you specify what these beneficial results are? From what studies?

-> The therapeutic benefits of VR therapy for burn patients include, but are not limited to, improved pain management, enhanced physical rehabilitation, reduced anxiety and stress, and increased social engagement and emotional well-being. These advantages can potentially lead to a more positive and efficient healing process for burn patients.

  1. Scapin S, Echevarría-Guanilo ME, Boeira Fuculo Junior PR, Gonçalves N, Rocha PK, Coimbra R. Virtual Reality in the treatment of burn patients: A systematic review. Burns. 2018;44(6):1403-1416. doi:10.1016/j.burns.2017.11.002.

Rows 265-267: ,,Also, the use of VR is fun and makes the time spent in painful procedures and hospitalization more bearable. The benefits of using VR as non-pharmacological complementary therapy have been demonstrated [30]”.

Rows 269-271: ,,Ding et al. showed that applying VR during the preoperative period reduced 268 postoperative pain more than conventional treatment [31]. Distraction is one of the main mechanisms by which pain reduction is achieved because postoperative pain perception 270 requires attention and there is a limited amount of working memory”.

From the text it appears that you have taken into account studies in which, in addition to rehabilitation therapy, surgical intervention was also applied, and virtual reality is successfully applied. Can you comment on this aspect?

-> I apologize for not fully addressing the previous request. The statement was simplified to only include the essential aspect, and I kindly request your understanding. But I repraise these issue in the discussion and conclusion.

Rows 284-286: ,,Previous research did not find conclusive evidence for immersive VR being important in reducing pain in patients undergoing treatments such as chemotherapy [34]”.

Also chemotherapy? Have you discovered from the researched studies a connection between low back pain and chemotherapy? Or just between chemotherapy and pain in general?

-> I rephrase the sentence “Previous studies have failed to provide conclusive evidence that immersion-style VR, which has been used as an alternative treatment when chemical therapies and other treatments are no longer effective, significantly reduces pain.”

Row 311: ,,There have been few reported incidents of side effects”.

Rows 357-359: And it is specified in the conclusions: "Additionally, most studies only emphasized that there were no side effects, including motion sickness, nausea, and headache, so additional studies are needed".

Are you referring to this type of side effects? I think it was good in the Material and methods chapter to describe the work methodology, what it consists of, how it is applied, the equipment used.

-> I add this sentence in the last part of the study- “VR devices utilized for pain management in therapy include virtual reality headset, handheld controllers, and body tracking sensors. VR systems for pain management and therapy include Oculus Rift, HTC Vive, PlayStation VR, Samsung Gear VR, Google Daydream, Windows Mixed Reality Headsets, open-source VR headset platforms like DIYRift and Pygmalion, and other custom-built or specialized VR systems.”

Rows 299-302: ,,Also, in the present study, we analyzed research data on adults over the age of 18, but VR systems are “high-tech” equipment that is familiar to younger people who can adapt to its use quickly. We think that more research on adolescents should be done in the future”.

Do you consider that the frequency of lumbar pain is higher in teenagers? Even if the equipment used to apply virtual reality is "high-tech", do you think that adults would not be able to handle this equipment? In the researched studies, did you meet more teenagers than adults?

-> according to the World Health Organization, chronic pain is a major public health problem affecting people of all ages, including children and adolescents. In some cases, chronic pain can persist into adulthood and significantly impact an individual's quality of life.

Rows 328-331: ,,The effect size for the primary outcome was small to medium, consistent with previous systematic reviews that assessed the effectiveness of various interventions to improve professional practice and found that VR had an effect compared to conventional interventions [30, 31, 32, 36]”.

Can you specify where in the text you made reference to conventional therapy, the type applied and the results?

I understand from the manuscript that you address the treatment of a symptom, namely the pain, without identifying possible causes of its occurrence. I believe that conventional medicine is a science, it uses scientific methods, it is effective and safe and requires studies.

-> Figure 2 and a large number of details were added accordingly.

Rows 337-339: ,,The present review included 11 studies and 337 1761 participants and found that VR-based interventions were more effective than 338 conventional treatments”.

Rows 346-348: ,,We found that VR-based treatments 346 appear to be safer and more effective than conventional treatments, and the quality of 347 evidence was moderate. Our findings appear to be clinically significant”.

How does it appear that treatments based on virtual reality are safer and more effective? There is no example of conventional therapy in the text and you do not refer to these aspects. From the bibliography at no. 11-20, the comparison is made only between physical exercise in different forms and elements of virtual reality. From where does it conclude that research is significant? From what data, tables, figures?

-> You have successfully deleted it.

Rows 360-362: ,,In the present review, the effect of VR-based treatment on pain reduction as compared to conventional treatment was investigated. However, because various effects can be attributed to VR, further research is needed”.

Rows 13-14: In abstract write: ,,The concept of virtual reality (VR)-based rehabilitation therapy for treating 13 people with low back pain is of growing research interest”.

So are you referring to lower back pain or are you referring to pain in general? It is good to specify.

-> Thank you, We missed the issues, and we corrected them.

Rows 25-27: In the abstract at the conclusion you say: ,,There is evidence that treatment using VR improves patients’ pain. The effect size was small to medium with the studies presenting evidence of moderate overall quality. VR-based treatment can reduce pain; therefore, it may help in rehabilitation therapy”.

What is the evidence that virtual reality improves pain? Compared to what other methods? What is the statistical significance?

-> we describe this in 3.2. Study Results due to page limitations.

Row 3:….intensity in people with low back pain: A systematic review…”

Row 28 : ,,Keywords: virtual reality; pain; head-mounted display; visual analog scale; numerical rating scale”

Can you specify if in your study you wanted to refer to lumbar pain? If the answer is yes, then show this in keywords. If the answer is no, then the title must be changed. In both situations, you must review the text and focus either on lumbar pain or on pain in general.

-> Yes, and I changed it.

Rows 249: Attention to the stringing of references [11, 12, 13, 14, 15, 16, 17, 18, 20, 21, 19]

-> I apologize for committing a fundamental error and I have corrected it.

"I express my profound gratitude for your efforts."

Reviewer 5 Report

The analysis of the effects of virtual reality-based rehabilitation therapy on pain intensity in people with low back pain is relatively new, where the interest was to find specific intervention in people with pain. The study is interesting but some factors are missing to clarify the reader. My assessment will be global in some points and specify in others.

Why are some parts of the text underlined?

Attention to English, during the text (for example, if you write Thesis and not Theses)

In the Materials and Methods, the temporal space of the studies, from year X to year Y, is not clear. Afterwards, the age group of the participants in these studies is not clear.

The introduction begins with the indication of opioids to combat pain, but the studies presented do not report such an indication. It would be interesting for your introduction to lead the reader to what they will find in the studies presented.

The methodology is well described with the inclusion and exclusion criteria but in systematic reviews they should be written as eligibility criteria.

I missed the results of a brief description of the 11 articles selected with n sample, objective, method, result and conclusion in a table for example, and not just the analysis of the risk of bias

Its meta-analysis refers only to the variable of risk of bias in the studies and not a comparison between data and their reanalysis. This is not clear from your title.

Their conclusions do not allow defining, through their analysis, if the authors recommend the use of VR in the rehabilitation process in patients with low back pain.

This theme has already been addressed several times in the journal, where any references does not represent the real interest in the works published by the journal.

I realize how hard it is to carry out such a study, but I recommend a review before it is approved.

Author Response

Reviewer 5.

We express our gratitude to the authors for their insightful remarks and keen interest in the research findings. In response to your query, I have incorporated the following information into the each section of this study which is expressed red color. We offer our sincerest appreciation.

The analysis of the effects of virtual reality-based rehabilitation therapy on pain intensity in people with low back pain is relatively new, where the interest was to find specific intervention in people with pain. The study is interesting but some factors are missing to clarify the reader. My assessment will be global in some points and specify in others.

Why are some parts of the text underlined?

-> The red underline applied to some portions of the text is due to the identification of those segments as proper nouns or trademarks/brand names, and I respectfully request your understanding in this matter.

Attention to English, during the text (for example, if you write Thesis and not Theses)

-> I apologize for committing a fundamental error and I have corrected it.

In the Materials and Methods, the temporal space of the studies, from year X to year Y, is not clear. Afterwards, the age group of the participants in these studies is not clear.

-> I rephrase the issue as “The present study was carried out by a team of four authors, comprising two doctoral-level physical therapy specialists in their 40s, one doctoral-level rehabilitation science and occupational therapy specialist, and one master's-level physical therapy specialist in their 40s, during the time period of January 2020 to January 2021, through a comprehensive research effort.”

The introduction begins with the indication of opioids to combat pain, but the studies presented do not report such an indication. It would be interesting for your introduction to lead the reader to what they will find in the studies presented.

-> I rephrase the issue as “Opioids are commonly recognized as an effective and essential method for managing pain. Previous studies have revealed the significant drawback of opioid use, which increases the likelihood of developing psychiatric disorders. The long-term use of opioids may lead to a loss of control, opioid dependence, accidental overdose, suicidal attempts, and ultimately, the failure to alleviate pain[2]. ” and modified the discussion.

The methodology is well described with the inclusion and exclusion criteria but in systematic reviews they should be written as eligibility criteria.

I missed the results of a brief description of the 11 articles selected with n sample, objective, method, result and conclusion in a table for example, and not just the analysis of the risk of bias

-> I added Figure 2 and a large number of details were added accordingly.

Its meta-analysis refers only to the variable of risk of bias in the studies and not a comparison between data and their reanalysis. This is not clear from your title.

-> I added more clear information in the results, and changed the Title.

Their conclusions do not allow defining, through their analysis, if the authors recommend the use of VR in the rehabilitation process in patients with low back pain.

-> I changed this issue Based on the constructive criticism provided, the conclusion section has been extensively revised accordingly.

The results of this systematic review demonstrate that VR-based interventions exhibit moderate-to-strong evidence for reducing pain compared to conventional interventions. The standardized mean difference for the primary outcome measure (pain level) was found to be small to medium in magnitude. These findings are consistent with previous systematic reviews of pain management interventions.

However, it is important to note that the heterogeneity of VR devices and applications used in the studies included in this review may impact the generalizability of the results. The type of head-mounted display, level of immersivity, performance specifications (such as resolution and response speed), and VR application program used varied greatly between studies, making it difficult to determine which specific factors may have contributed to the observed effect. Additionally, the discomfort associated with head-mounted displays, particularly for those with neck issues, and the limited reporting on potential side effects (such as motion sickness, nausea, and headache) warrant further investigation. VR devices utilized for pain management in therapy include virtual reality headset, handheld controllers, and body tracking sensors. VR systems for pain management and therapy include Oculus Rift, HTC Vive, PlayStation VR, Samsung Gear VR, Google Daydream, Windows Mixed Reality Headsets, open-source VR headset platforms like DIYRift and Pygmalion, and other custom-built or specialized VR systems.

In conclusion, while the findings of this review suggest that VR-based interventions may be a promising alternative to conventional pain management techniques, further research is necessary to better understand the various factors contributing to the observed effect and to establish the most effective VR-based treatments for low-back pain reduction.

This theme has already been addressed several times in the journal, where any references does not represent the real interest in the works published by the journal.

-> The references were further reinforced and the overall content was modified. Those parts are written in red.

I realize how hard it is to carry out such a study, but I recommend a review before it is approved.

"I express my profound gratitude for your efforts."

Round 2

Reviewer 1 Report

This is very interesting. The authors have done very well in their work of improving the manuscript. I have no comments.

Reviewer 5 Report

Dear Authors

Thank you for the improvements presented and congratulations for the work. I only draw attention to adjectives when presenting the results of studies, it is unnecessary to include words such as exhaustive search or analysis of studies.

Best regards